# Comparative transcriptome analyses in contrasting onion (*Allium cepa* L.) genotypes for drought stress

**Pranjali Ghodke**[1]*, **Kiran Khandagale**[1], **A. Thangasamy**[1], **Abhijeet Kulkarni**[2], **Nitin Narwade**[2], **Dhananjay Shirsat**[1], **Pragati Randive**[1], **Praveen Roylawar**[3], **Isha Singh**[4], **Suresh J. Gawande**[1], **Vijay Mahajan**[1], **Amolkumar Solanke**[5], **Major Singh**[1]

**1** ICAR-Directorate of Onion and Garlic Research, Rajgurunagar, Pune, India, **2** Department of Bioinformatics, Savitribai Phule Pune University, Pune, India, **3** S. N. Arts, D. J. M. Commerce and B. N. S. Science College, Sangamner, India, **4** School of Biomolecular Science, University College, Dublin, Ireland, **5** ICAR-National Institute for Plant Biotechnology, New Delhi, India

* pranjali.ghodke123@gmail.com

## Abstract

Onion (*Allium cepa* L.) is an important vegetable crop widely grown for diverse culinary and nutraceutical properties. Being a shallow-rooted plant, it is prone to drought. In the present study, transcriptome sequencing of drought-tolerant (1656) and drought-sensitive (1627) onion genotypes was performed to elucidate the molecular basis of differential response to drought stress. A total of 123206 and 139252 transcripts (average transcript length: 690 bases) were generated after assembly for 1656 and 1627, respectively. Differential gene expression analyses revealed upregulation and downregulation of 1189 and 1180 genes, respectively, in 1656, whereas in 1627, upregulation and downregulation of 872 and 1292 genes, respectively, was observed. Genes encoding transcription factors, cytochrome P450, membrane transporters, and flavonoids, and those related to carbohydrate metabolism were found to exhibit a differential expression behavior in the tolerant and susceptible genotypes. The information generated can facilitate a better understanding of molecular mechanisms underlying drought response in onion.

**Data Availability Statement:** All relevant data are within the paper and its Supporting Information files.

## Introduction

Bulb onion (*Allium cepa* L.) is an economically important vegetable crop cultivated worldwide in a diverse range of climatic conditions varying from temperate to semi-arid. India is one of the largest producers and exporters of onion globally. During 2017–2018, India produced 232 lakh tonnes of onion, of which 15.8 lakh tonnes was exported (http://agricoop.gov.in/). Asia contributes 67.5% of total world production, followed by Africa (12.9%), America (10.1%), and Europe (9.3%) (http://www.fao.org/faostat/en/#data/QC/visualize). However, drought stress causes approximately 30% yield losses in onion [1]. Stress due to biotic and abiotic factors is among the major constraints in exploiting the yield potential of the onion crop. In addition to biotic stress, onions are highly vulnerable to abiotic stresses such as extreme

**Funding:** This work was supported by ICAR-National Innovations on Climate Resilient Agriculture (NICRA)

**Competing interests:** The authors have declared that no competing interests exist.

temperature injuries, drought, and waterlogging [2, 3]. In India, the majority of onions are produced during the post-monsoon season. Being a shallow-rooted crop, in the post-monsoon season, onion is highly susceptible to mid-season drought due to low moisture resulting from inadequate rainfall and the shallowness of soil, which is insufficient to cater to the crop's water demand [4, 5]. Furthermore, the majority of available high-yielding modern onion varieties are developed for their best performance under optimum irrigation conditions. Therefore, genetic improvement of the existing genetic stock for drought tolerance is key to overcome the problem of drought-related yield losses in onion.

Drought tolerance is a complex phenomenon governed by numerous genes. Drought induces a vast array of plant responses that include a change in the gene expression pattern, accumulation of metabolites such as abscisic acid (ABA) or osmotically active compounds, and synthesis of specific proteins, namely largely hydrophilic proteins, oxygen radical scavenging proteins, and chaperones. Moreover, transcriptome analyses using microarray technology, along with conventional approaches, have identified many drought stress-responsive transcription factors (TFs) in plants [6, 7].

In recent years, plant transcriptome analysis using next-generation sequencing (NGS) technology has proven to be a robust and cost-effective tool for high-throughput sequence determination. NGS-based transcriptome data analysis facilitates differential gene expression (DGE) analysis at a global level, even when the plant genome sequence is unknown. This technology thus has been widely used in different economically important crops to identify DGE under various stresses; these genes are associated with different metabolic pathways and phenotypic traits [8–10].

In *Allium*, key metabolites and genes were identified through targeted metabolome and transcriptome profiling of dihaploid *A. cepa* and dihaploid *A. cepa* var. aggregatum under normal conditions and various stresses [11]. Han et al. [12] identified the genes that are differentially expressed during cold acclimation of onion genotypes and revealed the freezing tolerance mechanisms in onion crop. Zheng et al. [13] identified 39 CepNAC TFs (the NAC family of genes, particularly NAC-IV and NAC-V groups) that are likely to be involved in stress response in onion.

However, no study to date has determined the changes at the transcriptome level, DGE profiling, and alteration in the biochemical pathways of onion under drought stress. Therefore, in the present study, the transcriptome of two contrasting onion genotypes (accession nos. 1656 and 1627) subjected to drought stress was sequenced using Illumina paired-end sequencing technology. Due to the lack of the onion genome sequence and annotation information, the generated sequence data were *de novo* assembled to yield the transcriptome, which was then annotated using publicly available resources. To the best of our knowledge, this is the first report on the drought stress transcriptome of bulb onion. The study findings suggest the differential molecular behavior of the selected varieties toward drought stress. The study also provides a basis for elucidating the further understanding of transcriptional changes underlying the drought stress response in the onion crop.

## Materials and methods

### Plant material and drought treatment

The experiment was conducted at the ICAR-Directorate of Onion and Garlic Research (ICAR-DOGR), Pune, Maharashtra, India (N 18°84′, E 73°88′, H 553.8 m) under an automated rainout shelter. The identified drought-tolerant (Acc. 1656) and drought-senstive (Acc. 1627) onion genotypes were selected from the germplasm collection of ICAR-DOGR. Initially, seedlings were raised in the nursery on raised beds. Then, 6-week-old seedlings from

the nursery were transplanted in a plastic pot (height: 25 cm and diameter: 25 cm) of 12-kg capacity filled with field soil. The seedlings were raised under ambient growth conditions and irrigated at 100% field capacity until they reached the 5–6 leaf stage. Each treatment comprised 10 replicates (i.e., 10 pots/treatment). For drought stress treatment, 60% field capacity was maintained by withholding irrigation for 25–50 days after transplanting the seedling (drought stress-sensitive phase), and thereafter, normal 100% field capacity was retained. For the control treatment, 100% field capacity was maintained throughout the experiment. The recommended package of practices for onion was followed to raise a good crop. The leaf samples were harvested from each genotype for both the control and drought treatments. These samples were immediately frozen in liquid nitrogen and stored at −80 ˚C until use. The soil moisture level was monitored using the gravimetric method after every 24-h interval during the treatment. Additionally, to confirm the impact of drought stress, the plant water status was recorded by measuring relative water content (RWC) [14].

## Growth and physiological analyses

The growth of the plant was monitored by measuring plant height, number of leaves per plant, leaf area, and leaf length and width under the control and drought stress treatments at an interval of 6 days throughout the experiment. Total chlorophyll content of the onion leaves was estimated using the method described by Hiscox and Israelsta [15]. Total chlorophyll content was determined using the equation proposed by Arnon [16]. The membrane stability index (MSI %) was periodically measured (at 6-day interval) throughout the experiment by following the procedure described by Sairam et al. [17]. Total antioxidant capacity was estimated through ferric reducing antioxidant power (FRAP) assays, according to Benzie and Strain [18]. Total antioxidant capacity (TAC) was expressed as microgram ascorbic acid equivalents per milligram of fresh weight (FW). Total phenols were determined colorimetrically using the Folin-Ciocalteu reagent as described by Pinelo et al. [19]. The phenol content reported as gallic acid equivalent per gram FW of the sample. Proline accumulation was estimated according to the method given by Bates et al. [20]. The proline content was estimated from the standard curve using L-proline and expressed as μmol/g of FW.

## RNA isolation

Total RNA was isolated from the leaves of plants under the control and drought treatments in triplicate by using the modified CTAB and lithium chloride method [21]. The quantity and quality of RNA was determined using the NanoDrop® ND-1000 spectrophotometer (Thermo Fisher Scientific, USA) and agarose gel electrophoresis. The RNA samples were then treated with DNase I to avoid possible DNA contamination. Before cDNA synthesis, the integrity of RNA was determined using Bioanalyzer 2100 (Agilent technologies, Singapore). High-quality RNA having RIN values higher than 7 were pooled in equal quantities from three replicates of the control and drought-treated samples and were used for library construction and RNA-Seq analyses.

## Library preparation and RNA-Seq

The RNA samples that passed the quality check were used to prepare RNA-Seq paired-end sequencing libraries by using the Illumina TruSeq Stranded mRNA Sample Prep kit as per the manufacturer's protocol. In brief, the mRNA was enriched using poly-T beads and then fragmented enzymatically. First-strand cDNA synthesis was then performed using SuperScript II and ActD mix. The single-stranded cDNA was converted to double-stranded cDNA by using the second strand mix. The cDNA was then purified using AMpure XP beads, and poly(A)-

tailing, adaptor ligation, and enrichment were performed through PCR. The PCR-enriched libraries were analyzed in the Agilent 4200 TapeStation system (Agilent Technologies, USA) using a high-sensitivity D1000 Screen Tape as per the manufacturer's guidelines. The mean sizes of fragments in various libraries were 543, 524, 485, and 544 bp for 1656C, 1656D, 1627C, and 1627D, respectively. The libraries were then sequenced in the paired-end mode on Next-Seq500 using $2 \times 75$ bp platform chemistry.

### *De novo* transcriptome assembly and annotation

Quality of the captured high-throughput sequencing data was assessed using FastQC Toolkit v0.11.7 (https://www.bioinformatics.babraham.ac.uk/projects/fastqc/). The low-quality reads and adapters were removed from the raw reads using NGSQC Toolkit v2.3.3 (http://www.nipgr.ac.in/ngsqctoolkit.html) wherever necessary. The obtained high-quality raw reads ($\leq$Q20) were subjected to the Trinity assembler v2.2.0 (https://github.com/trinityrnaseq/trinityrnaseq/wiki) to build the merged transcriptome (control and treated) for the 1656 and 1627 genotypes in independent attempts. To obtain representative transcripts, we clustered the merged transcriptome using CD-HIT v4.7 (http://weizhongli-lab.org/cd-hit/download.php). The whole transcriptome quantitation was performed using Kallisto v0.44.0 (https://pachterlab.github.io/kallisto/download). DESeq, an R package, was used for DGE profiling. To assess DGE between two conditions, we used 2 as a log2fold change cut-off by selecting significant (p value $\leq$ 0.05) transcripts.

The final version of the transcriptome (CD-HIT clustered) was annotated using DIA-MOND BLASTX v0.8.22 (https://ab.inf.uni-tuebingen.de/software/diamond) utility against NCBI's non-redundant protein database (NRDB) (ftp://ftp.ncbi.nlm.nih.gov/blast/db/), Uni-Prot/SwissProt database (https://www.uniprot.org/), plantTFDB (http://planttfdb.cbi.pku.edu.cn/) with an e-value of $\leq 10^{-5}$. The gene ontology (GO) and pathway annotation fetched using the online UniProt/SwissProt ID mapping functionality. The Transeq utility is available under the EMBOSS v6.6.0 (http://emboss.sourceforge.net/download/) package used to convert the transcripts into the longest possible open reading frame. Such converted protein sequences were then scanned for the Clusters of Orthologous Group (COG) categories using a standalone version of emapper v1.0.3 against eggNOG v4.5.1 (http://eggnogdb.embl.de/#/app/downloads). The presence of various functional and/or conserved domains, protein families, and other important sequence signatures was determined by scanning the whole transcriptome against different databases such as Pfam, TIGRFAM, and SUPERFAMILY implemented under a standalone version of InterProScan v5.33–72.0 (https://www.ebi.ac.uk/interpro/download.html).

To study drought-responsive genes, we mapped the HQ reads on droughtDB (protein sequences) using the DIAMOND BLASTX utility with an e-value of $\leq 10^{-5}$ in an independent attempt for both samples of both genotypes. To capture the raw read count and coverage per gene, we used a pileup.sh program implemented under the BBMap suite (https://sourceforge.net/projects/bbmap/).

### Validation of DGE under drought stress using qRT-PCR

Transcripts that showed the differential expression behavior and encoded drought stress-responsive proteins were selected and validated using qRT-PCR. Transcripts such as methyl-malonyl-CoA epimerase, Ninja-family protein AFP3-like, vacuolar amino acid transporter, beta-galactosidase, WALLS ARE THIN1 (WAT1)-related protein, malate synthase, 21-kDa protein, NAC transcription factor 29-like, ABC transporter G (ABCG), protein STAY-GREEN, chaperone, L-ascorbate oxidase, superoxide dismutase, WRKY transcription factor

70, and aquaporin 1 were selected (S5 File). RNA was isolated from the same samples that were used for RNA-Seq analyses and treated with DNase I (Thermo Scientific, Lithuania) to remove possible DNA contamination. cDNA was synthesized using 1 μg of RNA from each sample using a cDNA synthesis kit (Thermo Scientific, Lithuania). Three biological and three technical replicates of each sample were used. Primers for the selected genes were designed using the Primer-BLAST program at NCBI (https://www.ncbi.nlm.nih.gov/tools/primer-blast/). The actin gene was used as an internal control of the experiment. Expression analyses of selected genes was performed in LightCylcer 480 II (Roche, Germany) using the LightCycler 480 SYBR Green I Master Mix kit (Roche, Germany). The relative expression and fold changes were calculated using the $2^{-\Delta\Delta Ct}$ method [22].

## Results

### Physiological and biochemical analyses

Physiological and biochemical parameters such as chlorophyll content, MSI, RWC, and anti-oxidant, phenol, and proline content were found to be higher in the tolerant genotype (1656) than in the susceptible genotype under drought stress (Fig 1; S1 File). The two genotypes significantly varied in plant height under drought stress. The tolerant genotype significantly maintained higher number of leaves and leaf area during stress treatment than the susceptible genotype (Fig 2), reflecting its ability to maintain higher photosynthesis activity under stressful conditions. RWC and MSI were directly proportional to drought tolerance and differed significantly among the studied genotypes as the stress increased. After 25 days of stress treatment, the tolerant genotype (1656) maintained higher plant RWC (>60%) and less membrane damage as reflected by their higher membrane stability (75%). Conversely, the sensitive genotype (1627) showed more membrane damage and lower tissue water content in response to water

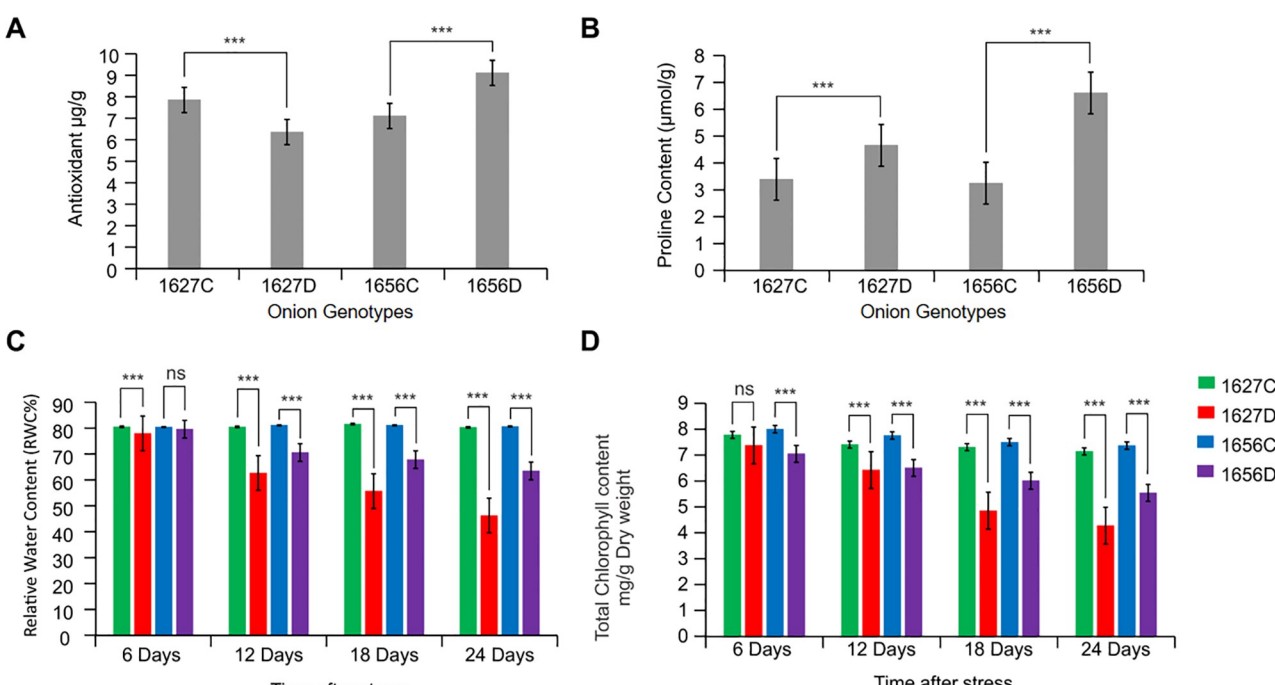

**Fig 1. Differential physiological and biochemical response in drought sensitive (1627) and tolerant (1656) onion genotypes.** A. Total antioxidant, B. Proline content, C. Relative water content and D. Total chlorophyll content.

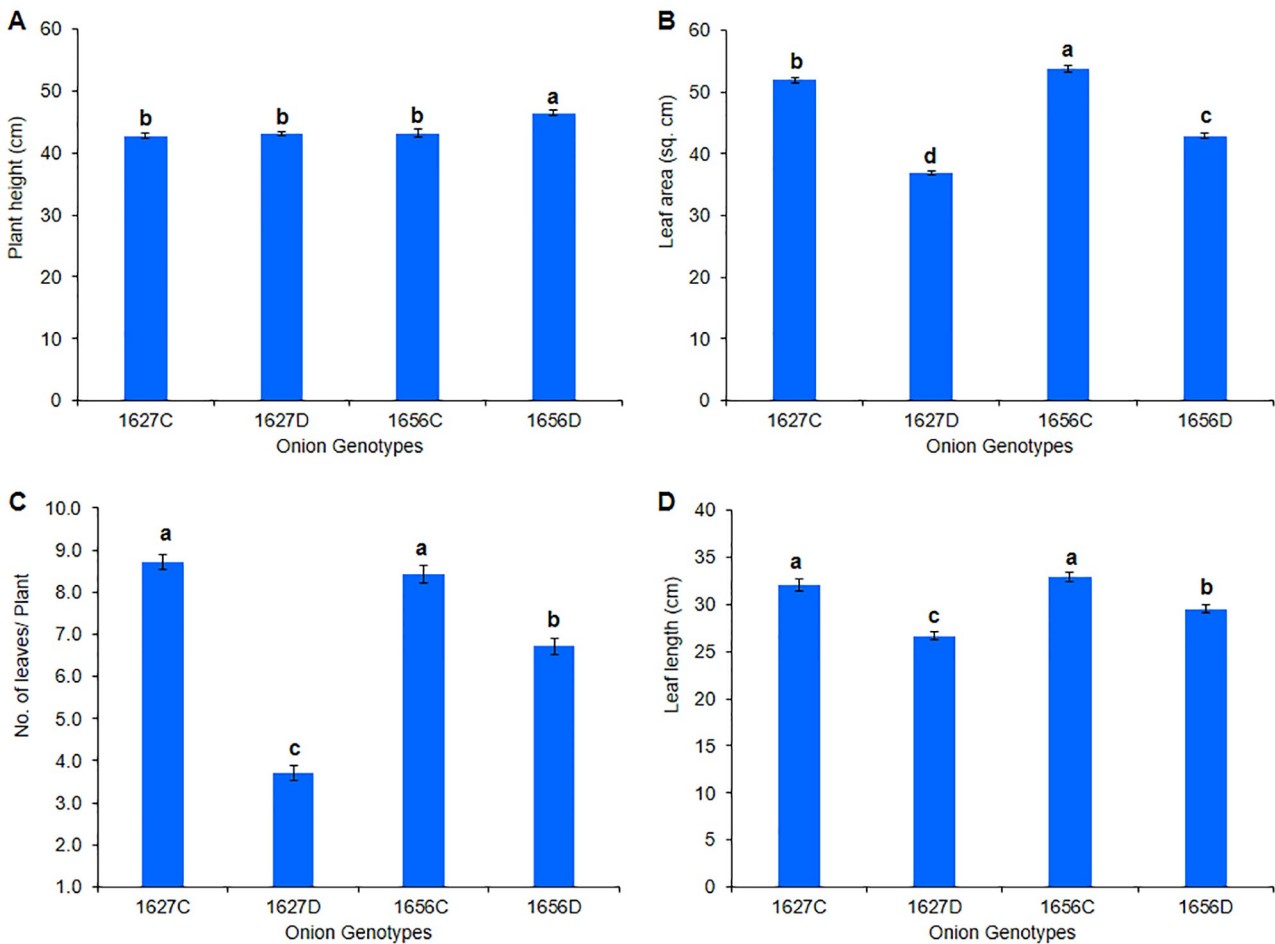

**Fig 2. Diffrential morphological response in drought sensitive (1627) and tolerant (1656) onion genotypes.** A. Plant height, B. Leaf area, C. Number of leaves, D. Leaf length.

stress. The leaf chlorophyll content also followed the same pattern, that is, it differed among the genotypes subjected to drought stress. The tolerant genotype retained significantly higher chlorophyll content as stress severity increased, whereas the leaf senescence rate became more pronounced in the susceptible genotype in response to water stress. Total phenol content, which is directly linked to onion pungency, was found to be elevated in response to drought stress. The tolerant genotype had 10% more phenol content than the susceptible genotype under water stress. Proline is a crucial drought stress indicator that plays a major role in cell osmoregulation. After 25 days of water stress treatment, proline content increased in the contrasting genotypes [204% in the tolerant genotype (1656) and 137% in the susceptible genotype (1627)]. This higher increase in the proline level reflects the drought adaptive mechanism present in the tolerant genotype (1656). Similarly, antioxidant enzyme activity, which is involved in scavenging reactive oxygen species (ROS) during oxidative or water stress, was found to be significantly higher in the tolerant genotype (1656) and lower in the sensitive genotype (1627). Thus, the tolerant genotype exhibited a drought adaptive mechanism that enabled it to survive in the water scarce environment.

**Table 1. Primary and final assembly statistics of 1656 and 1627 in control and drought stress treated onion libraries.**

| Parameters | 1656 | 1627 |
|---|---|---|
| Total No. of transcripts | 123206 | 139252 |
| Length of the transcriptome (Million Bases) | 84.57 | 96.29 |
| Max transcript length (bases) | 13837 | 15436 |
| Average transcript length (bases) | 686.48 | 691.50 |
| Median transcript length (bases) | 384.00 | 392.00 |
| N50 length (bases) | 1111 | 1107 |
| % GC | 37.77% | 37.57% |

## RNA-Seq and *de novo* transcriptome assembly

A total of 150.92 million raw reads generated from the control and drought-treated samples of drought-sensitive (1627) and drought-tolerant (1656) genotypes. In both the samples, we got sufficient HQ reads required for the transcriptome expression analysis, that is, on an average >30 million reads. The *de novo* transcriptome assembly yielded a total of 144668 and 164956 transcripts for samples 1656 and 1627, respectively. Then, these primary transcripts were clustered at 80% identity and 80% coverage cut-off to obtain the representative and non-redundant transcript set for both the samples in an independent attempt. A total of 123206 and 139252 non-redundant transcripts for 1656 and 1627, respectively, were clustered and resulted in the final transcriptome. The average transcript length was 690 bp with N50 statistics 1110 bp, and the maximum transcript length was 15436 bases while the GC content of the transcripts was 37.7% (1656) and 37.5% (1627) (Table 1). This set of transcripts was used for further downstream analysis. The raw sequencing data have been submitted to NCBI (BioProject: PRJNA595061).

## Differential gene expression

We performed *de novo* DGE analyses using aligned reads of the drought-tolerant (1656C vs 1656D) and drought-sensitive (1627C vs 1627D) onion cultivars. DGE analyses of 1656C versus 1656D resulted in the upregulation of 1189 genes and downregulation of 1180 genes, whereas in the case of 1627C versus 1627D, 872 genes were upregulated and 1292 genes were downregulated (Fig 3, S2 and S3 Files).

## Functional annotation of transcripts expressed in onion under drought stress

In total, 26428 (i.e., 21.45%) and 28109 (i.e., 20.18%) transcripts were successfully annotated with NCBI's non-redundant protein database (NRDB) from samples 1656D and 1627D, respectively, at an e-value of $\leq 10^{-5}$ and a query coverage of $\geq 50\%$. The reference database identifiers from the annotated transcripts were used for obtaining the GO and KEGG pathways along with other relevant information.

The transcripts were grouped into three categories based on the GO annotation: cellular components (CC), biological processes (BP), and molecular functions (MF). For sample 1627D, 255 unique CC categories were reported for 2252 transcripts (5.57%), whereas 3034 transcripts (7.51%) were annotated with 715 unique BP terms and 4864 transcripts (12.04%) showed hits against 757 unique MF terms. On the other hand, for sample 1656D, 255 unique CC categories were reported for 3442 transcripts (9.19%), 707 unique BP terms were reported

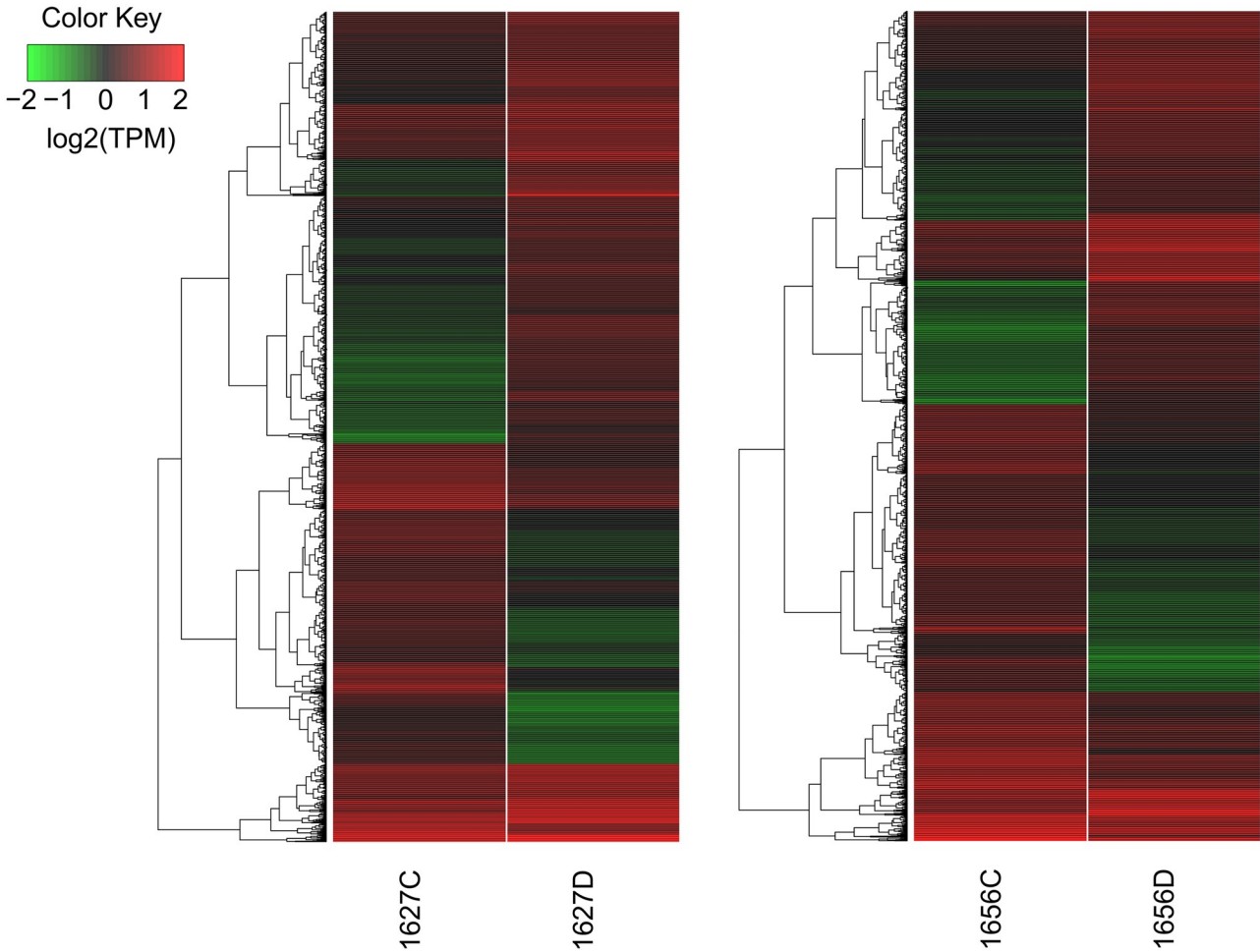

**Fig 3. Differential expression pattern showed by drought sensitive (1627) and tolerant (1656) onion genotypes under drought stress.**

for 4012 transcripts (10.77%), and 751 non-redundant MF terms were assigned to 6582 transcripts (17.68%). In both the samples, the MF terms were abundant, followed by the BP and CC categories. In the CC category, integral components of the membrane (GO:0016021) and nucleus (GO:0005634) were overrepresented, followed by the cytoplasm (GO:0005737), ribosome (GO:0005840), plasma membrane (GO:0005886), chloroplast (GO:0009507), and mitochondrion (GO:0005739). Similarly, the BP category indicates the high abundance of vital cellular processes such as photosynthesis, carbohydrate metabolism, cell wall organization, and transcription–translation. Among them, DNA integration (GO:0015074) and transcription (GO:0006451) were adequately represented. On the other hand, the MF category showed a high influence of MF such as nucleic acid binding, ATP binding, and protein kinase activity. In the MF category, nucleic acid binding (GO:0003676) and ATP binding (GO:0005524) were abundantly represented. (Fig 4A and 4B). In the KEGG pathway analysis, the highest percentage of the transcripts was reported to be involved in glycan metabolism (6%–7%), lipid metabolism (5%–6%), pectin degradation (4%–5%), and carbohydrate degradation (3%–5%) (Fig 4C and 4D). Of the total annotated transcripts, most transcripts matched with the *Asparagus officinalis* proteome (46%– 48%), followed by the proteomes of *Elaeis guineensis* (6.5%) and *Phoenix dactylifera* (5.3%) (Fig 4E and 4F).

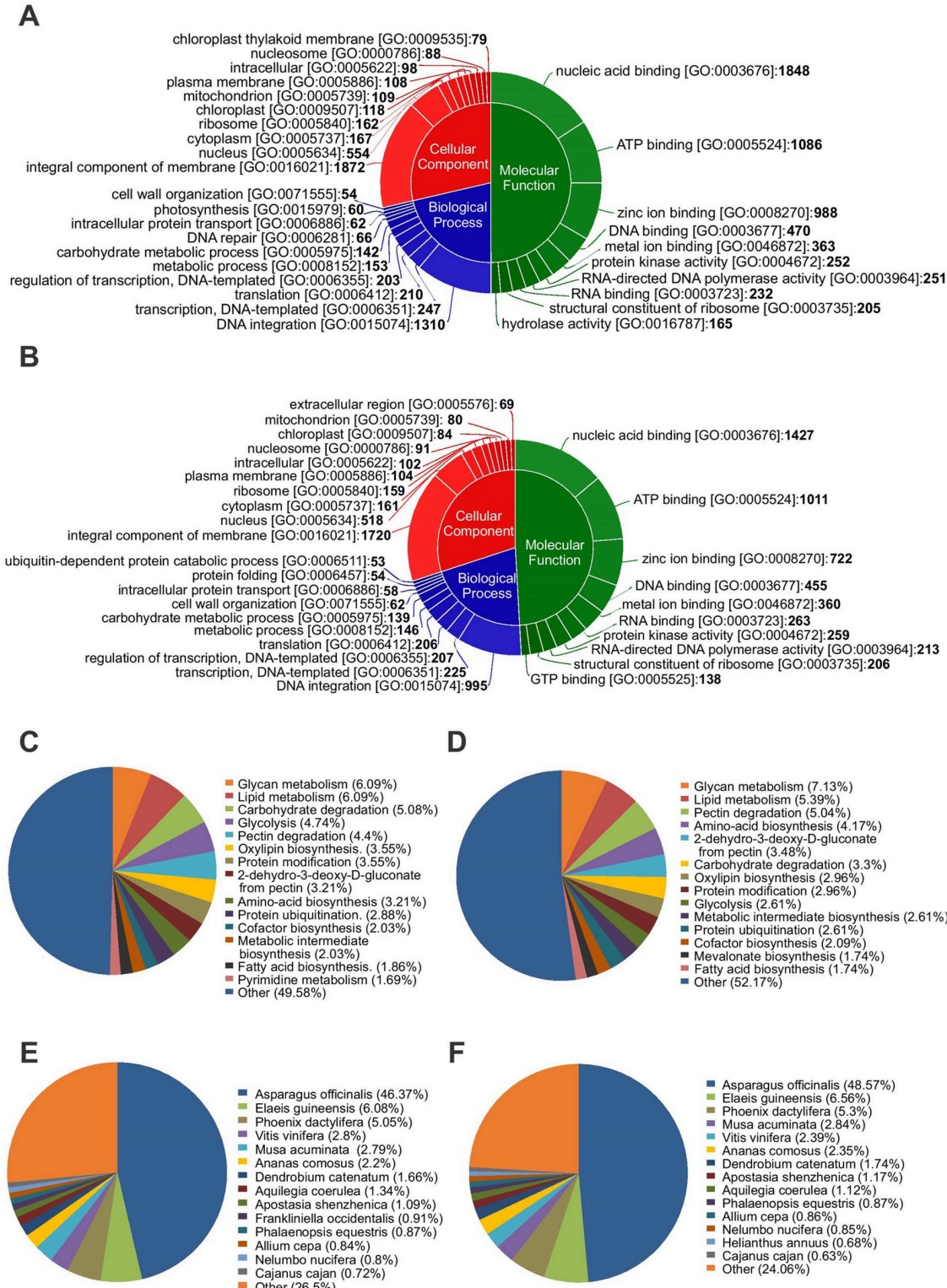

**Fig 4. Functional annotation of transcripts expressed in drought sensitive (1627) and tolerant (1656) onion genotypes under drought stress.** A, B: Gene ontology, C,D: KEGG pathway analyses, E,F: Species hit distribution.

The COG category distribution revealed that the major transcripts may contribute to signal transduction mechanisms (~7%), post-translational modification/protein turnover/chaperones (~6.8%), carbohydrate transport (4.92%), and metabolism/energy production and conversion (4.43%) COG categories in both the samples (S1 Fig, S4 File). The InterProScan analysis acknowledged the presence of various crucial conserved and functional signatures in our transcriptome such as membrane topology, signal peptides, and various functional domains (S1 Table). The homology-based sequence search against plantTFDB revealed the strong association of TFs extensively involved in plant growth promotion in both the samples such as MYB, bHLH, and ERF. The detailed distribution of TFs is presented in Fig 5. The aforementioned TFs were found to be upregulated in onion under drought stress.

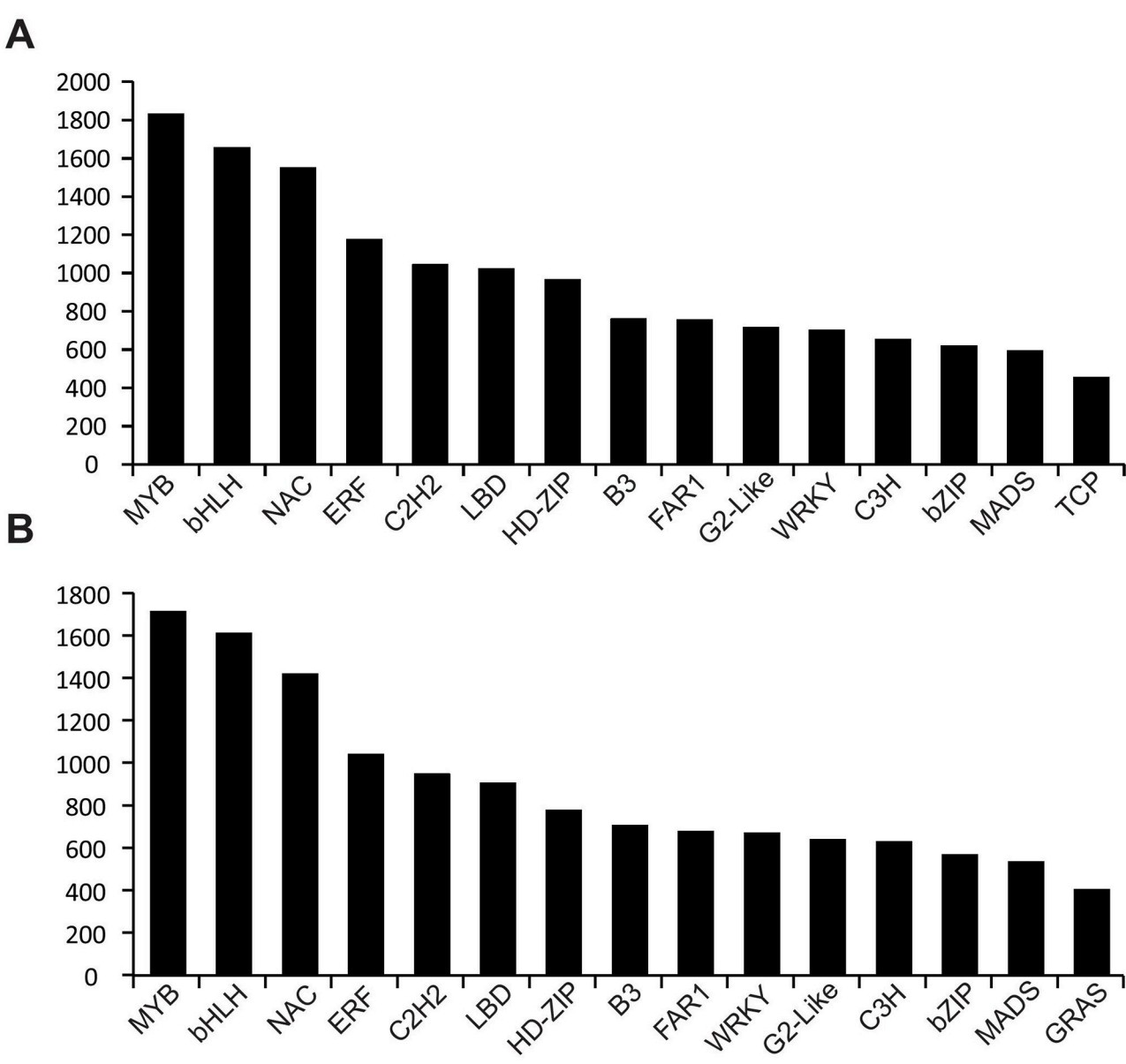

**Fig 5. Transcription factor distribution in assembled transcriptome of A. drought sensitive (1627) and B. tolerant (1656) onion genotypes under drought stress.**

## Drought-related gene expression analysis

The average gene sequence coverage ($\geq$50%) was considered to prune down the significant hits in the homology-based sequence search performed against droughtDB. The overall expression pattern of the significant genes from droughtDB was visualized in the form of a divergent plot by using fold change calculated by comparing the drought treatment with respective controls of the variety (Fig 6). TFs, membrane transporters, ABC transporters, cytochrome P450, antioxidants, and heat shock proteins were upregulated in onion cultivars under drought stress.

Among TFs, NAC, MYB, and WRKY families were highly upregulated in the tolerant cultivar. From 1656D, 10 members of NAC were differentially expressed and 9 were upregulated (up to 5.3-fold). While in the case of 1627D, 7 members of NAC were found to be differentially expressed and only 1 was upregulated. NAC29 was upregulated 4.8-fold in the tolerant cultivar (1656D). Similarly, in 1656D, of the 10 MYB family members, 8 were upregulated (up to 4-fold), whereas in 1627D, all 3 MYBs were downregulated in response to drought. WRKY TFs were also upregulated up to 5.5-fold in 1656D, whereas all WRKYs were downregulated in 1627D under drought.

Cytochrome P450 (CYP) genes were also found to be differentially expressed in response to drought stress in onion. CYP81, CYP71A, and CYP85A (7-, 6.1-, and 3.2-fold, respectively) showed high upregulation in the tolerant cultivar (1656) than in the susceptible cultivar (1627) (2.6-, −4-, and −2.5-fold, respectively). None of the P450 members were downregulated under drought stress in the tolerant cultivar, but several CYPs were downregulated in the susceptible cultivar.

Aquaporins occur in multiple isoforms in both plasmalemma and tonoplast membranes of plants. They regulate water transport in plants. In the present study, several aquaporins were differentially expressed in onion under drought stress such as aquaporin NIP1-1-like (5-fold) and aquaporin TIP3-2 (3.9-fold). Amino acid transporters such as vacuolar amino acid transporter (6.1-fold) and cationic amino acid transporter (3.6-fold) were upregulated in onion under drought stress. Similarly, the GABA transporter showed 4.8-fold upregulation in response to drought stress in onion.

ABA biosynthesis from carotenoids is catalyzed by 9-cis-epoxycarotenoid dioxygenase (NCED). A 3-fold increase was observed in the transcript level of *NCED* in the tolerant genotype (1656). Genes encoding ABA transporters such as the ABC subfamily G and NRT1/PTR family genes were also upregulated (3.9- and 6.2-fold, respectively) in the tolerant genotype. SNF1-related protein kinases were also upregulated (4.1-fold) in the tolerant genotype under drought stress.

Onion RNA-Seq data in the present study showed upregulation of flavonoid biosynthesis genes such as those encoding UDP-glycosyltransferase (4.9-fold), anthocyanidin 5,3-O-glucosyltransferase (3.2-fold), flavonoid glucosyltransferase (3.5-fold), and flavonol synthase (2.4-fold) in the drought-tolerant genotype (1656).

Moreover, genes for detoxification and ROS-scavenging enzymes (peroxidase, superoxide dismutase, and ascorbate oxidase) were found to be differentially expressed in onion under drought stress. Several genes related to carbohydrate metabolism, such as α-galactosidase (3.7-fold), β-galactosidase (7.3-fold), galactinol synthase (2.5-fold), galactinol—sucrose galactosyltransferase (4.9-fold), sucrose synthase (2.3-fold), and UDP-glucose 6-dehydrogenase (3.8-fold), were upregulated in the tolerant genotype (1656) under drought stress. Upregulation of these sugar metabolism genes in 1656 indicated their role in drought tolerance.

Overall, 21-kDA protein (9-fold), cytochrome P450 CYP81 (7-fold), RING-H2 finger protein (6.3-fold), momilactone A synthase-like (8.3-fold), peroxygenase 4 (6.7-fold), BAT-1

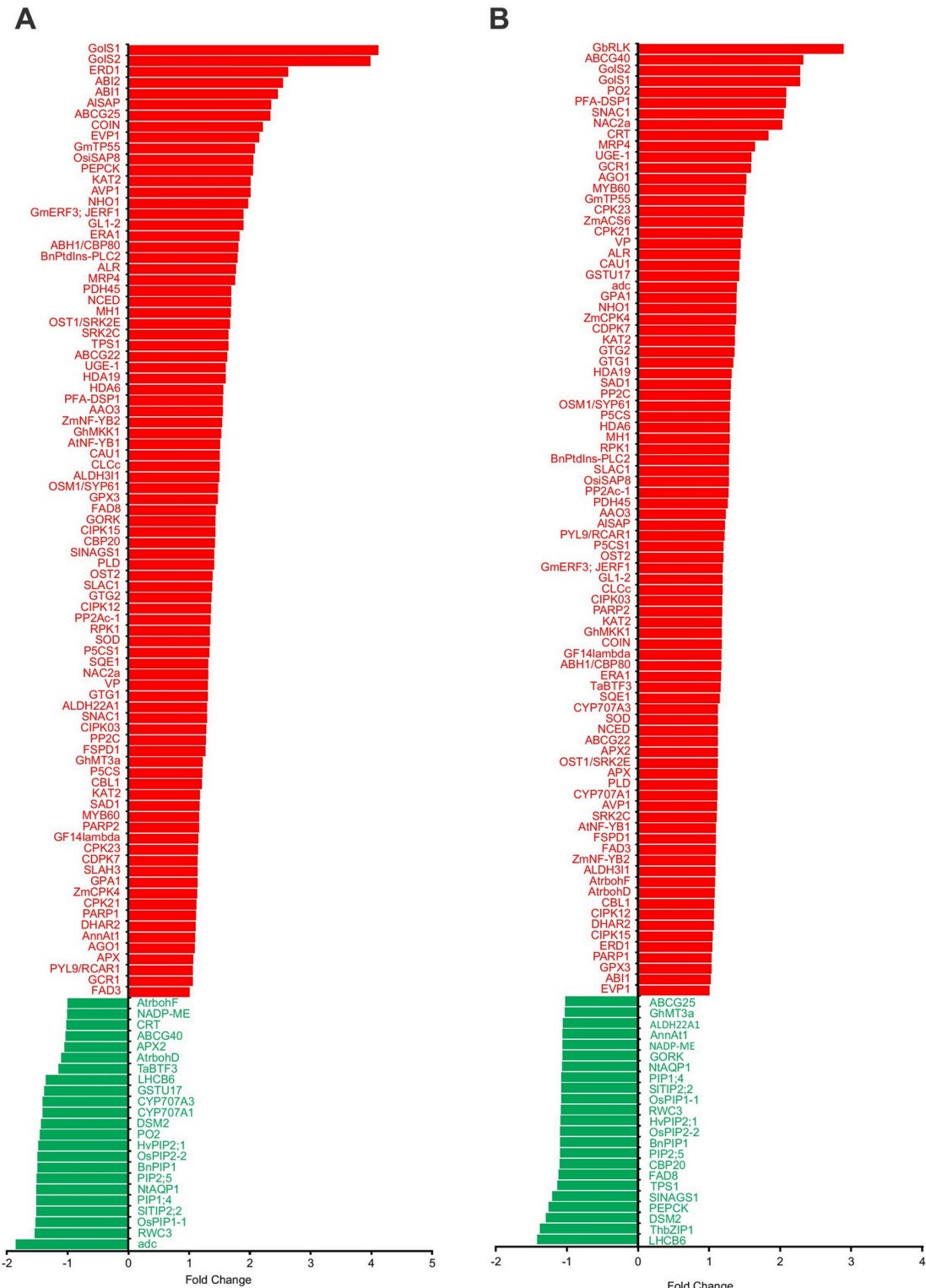

**Fig 6. Differential expression pattern (fold change) of the genes from droughtDB in A. drought sensitive (1627) and B. drought tolerant (1656) onion genotypes under drought stress.**

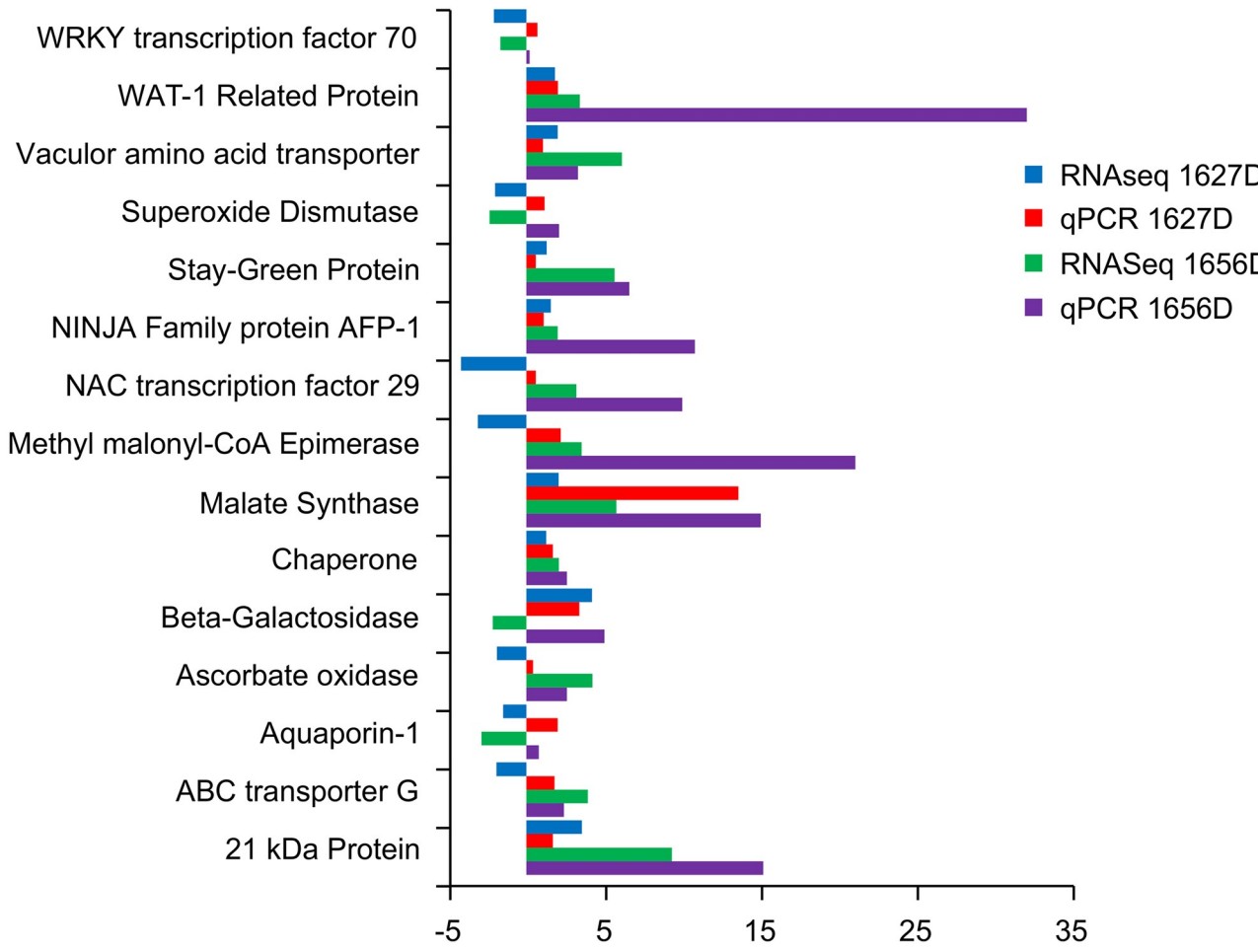

**Fig 7. Validation of few selected genes using qRT-PCR in A. drought sensitive (1627) and B. tolerant (1656) onion genotypes under drought stress.**

(DEAD Box BAT-1-like RNA helicase 15 isoform) (7.03-fold), and NAC29 (4.8-fold) are promising candidate genes that were upregulated manifold in the tolerant lines than in the sensitive counterpart in response to drought stress.

## Validation of DGE under drought stress using qRT-PCR

We validated the results of transcriptome analyses by using qRT-PCR of 15 randomly selected drought-related genes. The fold changes varied in RNA-Seq and qPCR analyses. However, the overall qPCR expression profile of most of the genes was in agreement with the RNA-Seq profile, which indicated the reliability of RNA-Seq data (Fig 7).

## Discussion

Two contrasting onion genotypes under drought stress were employed: drought-tolerant genotype (1656) and drought-susceptible genotype (1627). Their ability to tolerate drought stress for consecutive 25 days was evaluated on the basis of their morphological and biochemical performance. Being a shallow-rooted crop, onion requires frequent irrigation to maintain the desired yield and bulb quality [23]. According to a previous report, higher accumulation of

compatible solutes such as proline and soluble sugars and an increase in antioxidant enzyme activity play a substantial role in osmoregulation, thus improving cellular turgidity and membrane stability in the tolerant wheat cultivar under water stress [24]. Similar results were recorded in our earlier study with the well-known short-day onion cultivar Bhima Kiran, where the decline in overall physiological and biochemical parameters was recorded in response to drought stress, which affected the entire plant growth and yield [2]. Wakchaure et al. [25] reported that limited irrigation or water deficit stress in onion severely affects the crop growth rate as indicated by a reduction in plant height, leaf area index, and chlorophyll content and other important physiological parameters contributing to the overall bulb quality and yield.

## RNA-Seq and functional annotation

RNA-Seq is rapid, inexpensive, and independent of genome complexity, and thus, NGS has emerged as a method of choice for expression analyses, discovery of new genes, and development of molecular markers in crops where genome sequence information is not available. In the present study, we used the Illumina Next 500 platform and generated 150.92 million raw reads from the control and drought-treated samples of drought-sensitive (1627) and drought-tolerant (1656) genotypes. Similar transcriptome statistics were also reported in other RNA-Seq analyses in onion; Zhang et al. [13] generated 72.53 million 100-bp paired-end reads, while Shemesh-Mayer et al. [26] sequenced six libraries from 100-bp one-end reads. These RNA-Seq data suggested that *de novo* assembly was effective and captured the majority portion of the onion transcriptome.

For the functional annotation, we searched the final transcriptome against NCBI's NRDB using DIAMOND BLASTX utility. Very few curated protein sequences are available for *A. cepa* in the database; therefore, we got the annotation across various organisms. Of all other organisms, we found *A. officinalis* as the major contributor. Recently, Mehra et al. [27] performed a transcriptome analysis of snow mountain garlic and reported the highest homology with *A. officinalis*. Han et al. [12] conducted a similar study investigating the effect of cold stress on onion by using the transcriptome of *A. fistulosum*. Sun et al. [28] reported the highest match with *Vitis vinifera*, while Zhang et al. [13] reported the highest match with *E. guineensis*, followed by *P. dactylifera*. *A. officinalis* is closely related to the alliums, and the recent availability of its genome sequence resulted in the highest similarity with onion transcriptome data. Several TF families (MYB, WRKY, NAC, and bHLH) were identified by searching plantTFDB. These TFs are known to play a role in molecular regulation in response to biotic as well as abiotic stresses in several plants [29], including onion under the cold [12] and heat stress [30]. InterProScan analysis was performed for functional analysis of transcripts by classifying them into families and predicting domains and important sites. Transmembrane domains, conserved domains, and protein superfamilies were predicted from the present transcriptome data by using InterProScan. Similar analyses were performed for domain and motif information in garlic [27] and giant reed [31]. Almost 30% of transcripts in the present study were unknown or not annotated, and they might be unique to onion.

## Drought-related gene expression analysis

NAC29 is known for its role in abiotic stress tolerance, and transgenic *Arabidopsis* expressing NAC29 from wheat showed increased tolerance to salinity and dehydration stress by delaying senescence [32]. When overexpressed in *Arabidopsis*, MYB44 helps in ABA-mediated stomatal closure upon salt and drought stress [33]. MYB108 was reported to be highly upregulated under drought stress in poplar [34]. MYB108 and MYB39 were also upregulated in response to

heavy metal stress [35]. WRKY41 increased tolerance to salinity and drought stress when expressed in transgenic tobacco [36]. Similarly, along with other WRKYs, WRKY75 is known to modulate abiotic stress response [37] and is a modulator for phosphate uptake and root development in *Arabidopsis* [38]. Such a significant difference in the expression pattern of NAC, MYB, and WRKY suggests their crucial role in molecular reconfiguration at the RNA level, which ultimately imparted drought stress tolerance to 1656.

CYPs are a large and diverse family of genes in plants; several P450 genes are reported to increase abiotic stress tolerance [39]. CYP85A is involved in brassinosteroid biosynthesis, and overexpression of spinach CYP85A in tobacco led to an increase in drought tolerance as well as root development [40]. The upregulation of several CYPs was also reported in the transcriptome data of perennial ryegrass in response to heat stress [41]. Thus, the upregulation of P450 genes in 1656 might contribute to its drought tolerance.

NIPs have an essential role in maintaining water balance during drought and salinity stress [42]. RNA-Seq analyses of potato revealed the upregulation of aquaporin TIP3-2 under drought stress [43]. Thus, aquaporins can be considered potential drought tolerance-inducing proteins in onion and other *Allium* crops. Drought tolerance is partially associated with amino acid accumulation [44]. These amino acids serve as osmolytes, and ROS-scavenging and signaling molecules in plant stress response [45]. GABA is a well-known molecule involved in enhancement of abiotic stress tolerance [46, 47]. Moreover, few ABC transporters were upregulated under drought stress in onion. They are involved in stomatal closure during water stress.

ABA is known as a stress hormone because of its central role in response to various stresses. To cope up with stress with the help of ABA signaling, transcriptional upregulation of *NCED* occurs under drought stress [48]. ABA is generally transported by passive diffusion to guard cells. However, ABA can be transported by a few transporters such as members of the ABC subfamily G and NRT1/PTR family [49, 50]. They might facilitate stomatal closure in 1656 and assist in minimizing water loss. SNF1-related protein kinases are the subfamily of serine/threonine kinases that play a crucial role in ABA and sugar signaling [51]. SNF1-related protein kinases that are upregulated in onion under drought stress might phosphorylate the various TFs by regulating the ABA-dependent signaling cascade to enhance drought tolerance.

Flavonoids have antioxidant properties and are known to have a role in conferring abiotic stress tolerance to plants [52, 53]. Integrated RNA-Seq and metabolomics studies have revealed the upregulation of flavonoids and flavonoid biosynthesis genes in shallot doubled haploid [11].

Carbohydrate metabolism also plays a vital role in abiotic stress tolerance [54, 55]. Several sugar metabolism-associated genes were upregulated in onion under drought stress. These genes were reported to be upregulated in shallot doubled haploid and might have imparted stress tolerance to shallot than to onion [11]. Carbohydrate metabolism-associated genes were also upregulated in onion under cold stress [12] and in *Camellia sinensis* under drought stress [9].

Few important genes in abiotic stress response such as 21-kDa protein, cytochrome P450 CYP81, RING-H2 finger protein, momilactone A synthase-like, peroxygenase 4, and BAT-1 (DEAD Box BAT-1-like RNA helicase 15 isoform) were upregulated several fold in 1656 under drought stress. DEAD-box RNA helicases are proteins of a category that play a crucial role in maintaining cell genome integrity during stress conditions [56]. OsBAT1 was upregulated under abiotic stress in rice and showed unique characteristics such as unwinding of both DNA and RNA duplexes; bipolar translocation and its transcript upregulation under abiotic stresses, indicated that it is a multifunctional protein [57]. ROS signaling plays a critical role in plant responses to abiotic stress such as drought and salinity. CYPs are associated with protection of

plants from harsh environmental conditions by increasing the activity of compounds such as flavonoids that have an increased antioxidant activity [58]. However, the cytochrome P450 gene cluster member *TaCYP81D5* conferred salinity tolerance in wheat by ROS scavenging [59]. Peroxygenases are invloved in oxylipin metabolism and are important in plant stress response. In response to drought stress, peroxygenase 4 was upregulated (10.1-fold) in creeping bent grass (*Agrostis stolonifera*) [60]. Furthermore, RING-H2 finger proteins are a special type of zinc finger proteins known to increase stress tolerance by modulating the hormonal profile of tomato [61] and *Arabidopsis* [62] to cope up with adverse environmental conditions. Momilactones are allopathic phytoalexins involved in disease and weed resistance in rice. Xuan et al. [63] reported that momilactone A was more efficient in conferring salinity and drought stress tolerance than resistance to weed. Momilactone need to be characterized in onion for a better understanding of their role in conferring stress tolerance.

## Validation of DGE under drought stress using qRT-PCR

The expression trends of genes from qRT-PCR corresponded with those of transcriptome analyses, thus validating the RNA-Seq data. NAC29 from wheat enhanced drought and salt tolerance in *Arabidopsis* by delaying senescence and boosting primary root elongation [32]. STAY-GREEN is a well-known protein involved in drought tolerance in a number of crops; it acts by slowing down chlorophyll degradation [64]. In the present experiment, chlorophyll content in the drought-tolerant genotype (1656) was more under drought stress than in its susceptible counterpart (1627). This might be linked to the upregulation of the aforementioned genes. Ascorbate oxidases are involved in plant stress tolerance through ROS scavenging [65], which is upregulated in the drought-tolerant genotype (1656). Malate synthase, a marker for the glyoxylate cycle, was upregulated >15-fold under drought stress in the drought-tolerant genotype (1656). Malate synthase is also found to be upregulated by ABA treatment [66]. The 21-kDa protein was one of the prominent proteins to be upregulated in salinity stress in finger millet [67]. However, in the current study, the 21-kDa protein gene was upregulated 9.33-fold in the drought-tolerant genotype under drought stress. WAT1 is a vacuolar auxin transport facilitator required for auxin homoeostasis during drought stress [68]. WAT1 was upregulated in the drought-tolerant line of maize [69]. The TF ABA-Insensitive 5 (ABI5) is a key regulator of ABA signaling and stress response. ABI5-binding proteins are induced by ABA and/or dehydration stresses in *Arabidopsis* [70, 71]. ABA biosynthesis is highly induced by dehydration in the vascular parenchyma cells of roots and shoots. However, the plant ABCG was shown to transport terpenoids, and because ABA is a tetraterpene-derived sesquiterpene, ABCG proteins are strong candidates for ABA transporters [72].

## Conclusion

Onion is a shallow-rooted plant, and thus is likely susceptible to drought stress. Here, we performed transcriptome sequencing of drought-tolerant and susceptible onion genotypes. More than 1100 differentially expressed genes were identified from these genotypes under drought stress. These genes were functionally annotated using various standard bioinformatics programs. Several drought-responsive genes were upregulated in the tolerant genotype (1656) such as those encoding TFs, cytochrome 450, and membrane transporters, and those associated with carbohydrate metabolism and flavonoid biosynthesis. These genes might confer drought tolerance in this onion genotype at the molecular level. Physiological and biochemical parameters also indicated the better performance of the 1656 genotype over the 1627 genotype under drought stress conditions. To our best knowledge, the present study is the first to report the transcriptomic analysis of drought response in onion. The study findings will help

researchers have an improved understanding of the molecular basis of drought response in onion.

## Supporting information

**S1 Table. InterProScan analysis.**
(DOCX)

**S1 Fig. COG analyses of A: 1627; B: 1656.**
(TIF)

**S1 File. Data of physiological and biochemical analyses.**
(XLSX)

**S2 File. Differential gene expression of 1627.**
(XLSX)

**S3 File. Differential gene expression of 1656.**
(XLSX)

**S4 File. GO analyses.**
(XLSX)

**S5 File. Primers used for qRT-PCR.**
(XLS)

## Author Contributions

**Conceptualization:** Pranjali Ghodke, Suresh J. Gawande, Vijay Mahajan.

**Data curation:** Kiran Khandagale, Dhananjay Shirsat.

**Formal analysis:** Kiran Khandagale, A. Thangasamy, Abhijeet Kulkarni, Nitin Narwade.

**Funding acquisition:** Pranjali Ghodke, A. Thangasamy.

**Methodology:** Kiran Khandagale, Dhananjay Shirsat, Pragati Randive, Praveen Roylawar.

**Project administration:** Pranjali Ghodke.

**Resources:** Suresh J. Gawande, Major Singh.

**Software:** Nitin Narwade, Praveen Roylawar, Isha Singh.

**Supervision:** Pranjali Ghodke, Suresh J. Gawande, Major Singh.

**Validation:** Kiran Khandagale.

**Visualization:** Abhijeet Kulkarni, Nitin Narwade.

**Writing – original draft:** Kiran Khandagale, Abhijeet Kulkarni, Dhananjay Shirsat, Pragati Randive.

**Writing – review & editing:** Kiran Khandagale, Suresh J. Gawande, Amolkumar Solanke.

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
