## [Decision Letter · Decision Letter 0]

26 May 2020

PONE-D-20-06651

Comparative transcriptome analyses in contrasting onion (Allium cepa L.) genotypes for drought stress

PLOS ONE

Dear Dr. G,

Thank you for submitting your manuscript to PLOS ONE. After careful consideration, we feel that it has merit but does not fully meet PLOS ONE’s publication criteria as it currently stands. Therefore, we invite you to submit a revised version of the manuscript that addresses the points raised during the review process.

ACADEMIC EDITOR: The reviewers make critical recommendations, but an important consideration is whether the study is technically sound and describes a significant new advance in the area. Unfortunately it appears that additional work is needed as indicated by reviewers. If these were meticulously performed, then I am sure that the MS could be reconsidered.

Manuscript require improvement in grammar, usage, and overall readability

We look forward to receiving your revised manuscript.

Kind regards,

Kundan Kumar, PhD

Academic Editor

PLOS ONE

Reviewers' comments:

Reviewer's Responses to Questions

**Comments to the Author**

1. Is the manuscript technically sound, and do the data support the conclusions?

Reviewer #1: Yes

Reviewer #2: Yes

2. Has the statistical analysis been performed appropriately and rigorously? 

Reviewer #1: Yes

Reviewer #2: Yes

3. Have the authors made all data underlying the findings in their manuscript fully available?

Reviewer #1: Yes

Reviewer #2: Yes

4. Is the manuscript presented in an intelligible fashion and written in standard English?

Reviewer #1: Yes

Reviewer #2: No

5. Review Comments to the Author

Reviewer #1: Though RNA-Seq based works are very common now-a-days for investigating role, discovery of candidate genes and markers and differential gene expression profiling, yet the work entitled “Comparative transcriptome analyses in contrasting onion (Allium cepa L.) genotypes for drought stress” is important in two respects, one the crop is commercially important and second, drought stress which causes huge yield loss. I commend the authors for this study, and would like some clarity on following points:

Comment 1:

“Authors performed de novo DGE analyses using aligned reads of drought-tolerant (1656C vs 1656D) and drought-sensitive (1627C vs 1627D) onion cultivars”.

I believe, the comparative analysis should have been made also between 1627D vs 1656D and 1627C vs 1656C. This could be important in better understanding the drought stress in tolerant and susceptible genotype.

Comment 2:

Why not genes were selected based on their high fold change value (negative and positive) and drought-related instead of random selection just for validation of RNA-Seq identified DGEs. Also the gene expression analysis at 6, 12, 18, 24 days should have been performed to correlate with biochemical and physiological data related to drought stress.

Comment 3:

The study comes out with a long list of drought stress-related such as genes encoding transcription factors, cytochrome P450, membrane transporters, flavonoids, and carbohydrate metabolism, etc. which showed differential expression behavior in tolerant and susceptible genotypes, it would not be much helpful in understanding the key role players. Can authors enlist few candidate genes for drought stress in onion based on present study?

Comment 4:

“Total phenol content that directly linked with the onion pungency and found to be elevated in response to drought stress”. Why none of the genes related to phenylpropanoid/Flavonoid pathway were included qRT-PCR.

Comment 5:

Discussion lacks on up-regulated genes in tolerant genotype like WAT 1 related protein, NINJA family AFP1, Methyl malonyl co-A epimerase and 21KDa Protein.

Comment 6:

Some figures labels needs clarity such as Figure 3 and Figure 5, the labels are difficult to read.

Reviewer #2: Introduction about onion is very shallow. Provide the data of onion production and loss occurs due to other stress and then mention the yield loss due to drought.

Introduction line no. 70: mention the name of the genotypes.

Introduction line no. 75-80 is the part of the result and do not need to describe here.

MM: 89: 1656 and 1627 are the cv nos? specify clearly.

MM: irrigated at 100% field capacity until they reached the 5-6 leaf stage, how come it is mentioned as field capacity, though plants are in pots.

L188: mention the name of selected genes, and site the table for the primers here.

Fig1 A: change sample as genotypes and treatments.

Morphological data should be added in main figure, shift from supplementary to main text.

L 200: significantly maintained the number of leaves, is it higher then edit the sentence as

Significantly maintained higher number of leaves

For membrane damage, authors are advised to include MDA data.

L 208, 209: incomplete sentence, please edit it: The observation recorded for leaf chlorophyll content was also in the same line 209 differing among the genotypes subjected to drought stress.

L 212: Total phenol content that directly linked with the onion pungency and found to be

elevated in response to drought stress.: and should be was

L258: , needs to be place before and after respectively.

Fig 3A-F, Fig. 5: font size is too small and is not readable.

L 305-353: data mentioned in the result section should be properly checked as at some point the fold expression has not been mentioned. Authors should uniformly mention the fold expression, where it upregulated or down regulated

L 354-359: The real time expression in the result section is written as MM part and result is missing. Pl write the result part properly. Move this write up in MM.

Prediction of SSR data does not fit relevant.

The authors are advised to correct the MS thoroughly.

6. PLOS authors have the option to publish the peer review history of their article (what does this mean?). If published, this will include your full peer review and any attached files.

Reviewer #1: Yes: Dr. Ravi Shankar Singh

Reviewer #2: Yes: Pradeep K Agarwal

---

## [Author Response · Author response to Decision Letter 0]

20 Jul 2020

As per Editor's suggestion the manuscript copyedited from "Scholarly Editing and Translation Services Pvt. Ltd." for language usage, spelling and grammar. Also we ensured that the manuscript meets PLOS ONE's style requirements, including the file names.

The reviewer's comments are responded as following.

Reviewer #1: Though RNA-Seq based works are very common now-a-days for investigating role, discovery of candidate genes and markers and differential gene expression profiling, yet the work entitled “Comparative transcriptome analyses in contrasting onion (Allium cepa L.) genotypes for drought stress” is important in two respects, one the crop is commercially important and second, drought stress which causes huge yield loss. I commend the authors for this study, and would like some clarity on following points:

Comment 1:

“Authors performed de novo DGE analyses using aligned reads of drought-tolerant (1656C vs 1656D) and drought-sensitive (1627C vs 1627D) onion cultivars”. I believe, the comparative analysis should have been made also between 1627D vs 1656D and 1627C vs 1656C. This could be important in better understanding the drought stress in tolerant and susceptible genotype.

Response: We do agree with this comment. We would be happy to perform additional analysis suggested by the reviewer. But, due to Covid-19 pandemic, our research institute and Pune University are not being fully functional. We are in hotspot zone, our access to labs have been restricted. The present lockdown has been extended till 31st July, 2020, since positive cases are going up substantially, lockdown is most likely extended beyond 31st July. Therefore, we could not able to perform the suggested additional analysis. However, we feel that the analyses in the submitted manuscript are adequate enough to support the proposed study. Further, we accept the suggestion and we are planning to publish the detailed comparative analysis of 1627D vs. 1656D and 1627C vs. 1656C as a sequel to the present manuscript as and when things gets normal.

Comment 2:

Why not genes were selected based on their high fold change value (negative and positive) and drought-related instead of random selection just for validation of RNA-Seq identified DGEs. Also the gene expression analysis at 6, 12, 18, 24 days should have been performed to correlate with biochemical and physiological data related to drought stress.

Response: The purpose of the qRT-PCR in the present study was only to validate the transcriptomics data. Most of the transcripts we selected for qRT-analysis were directly linked with the drought stress. We already have performed biochemical analyses of drought related markers at different time points (6, 12, 18, 24 days). Many of the genes showed high fold change values were linked to these markers. The additional gene expression analysis would not add any additional information to the manuscript. The suggested analysis would have strengthened the validation only that we have adequately performed in the manuscript.

Comment 3:

The study comes out with a long list of drought stress-related such as genes encoding transcription factors, cytochrome P450, membrane transporters, flavonoids, and carbohydrate metabolism, etc. which showed differential expression behaviour in tolerant and susceptible genotypes, it would not be much helpful in understanding the key role players. Can authors enlist few candidate genes for drought stress in onion based on present study?

Response: Yes, we have listed few genes which showed multi-fold upregulation under drought stress in result section and their supporting discussion was incorporated in revised MS.

Comment 4:

“Total phenol content that directly linked with the onion pungency and found to be elevated in response to drought stress”. Why none of the genes related to phenylpropanoid/Flavonoid pathway were included qRT-PCR.

Response: Thanks for the suggestion; we already performed total phenol analysis that is strongly linked with the pungency. The purpose of the qRT-PCR in the present study was only to validate the transcriptomics data. Therefore, we randomly selected genes for qRT-PCR to validate the RNAseq data without targeting any pathway. However, we will certainly target the Phenylpropanoid/Flavonoid pathways for further probing into the drought mechanisms in our future studies.

Comment 5:

Discussion lacks on up-regulated genes in tolerant genotype like WAT 1 related protein, NINJA family AFP1, Methyl malonyl co- A epimerase and 21KDa Protein.

Response: As per suggestion, we have added discussion on suggested genes in revised MS. 

Comment 6:

Some figures labels needs clarity such as Figure 3 and Figure 5, the labels are difficult to read.

Response: Figure 3 and 5 are changed in revised MS. 

Reviewer #2: 

Introduction about onion is very shallow. Provide the data of onion production and loss occurs due to other stress and then mention the yield loss due to drought.

Response: As per suggestion of reviewer, we have added data in revised MS. 

Introduction line no. 70: mention the name of the genotypes.

Response: Names of genotypes were mentioned in revised MS

Introduction line no. 75-80 is the part of the result and do not need to describe here

Response: Suggested part is deleted in revised MS

MM: 89: 1656 and 1627 are the cv nos? specify clearly.

Response: These are accession number of genotypes under study. It is specified in revised MS. 

MM: irrigated at 100% field capacity until they reached the 5-6 leaf stage, how come it is mentioned as field capacity, though plants are in pots.

Response: These plants were watered regularly as per the crop requirement so that they do not feel water stress 

L188: mention the name of selected genes, and site the table for the primers here.

Response: Changes made as per suggestion in revised MS. 

Fig1 A: change sample as genotypes and treatments.

Response: Changes made in figure as per suggestion 

Morphological data should be added in main figure, shift from supplementary to main text.

Response: Morphological data is added in main manuscript as Figure 2. 

L 200: significantly maintained the number of leaves, is it higher then edit the sentence as

Significantly maintained higher number of leaves

Response: Sentence modified as per suggestion

For membrane damage, authors are advised to include MDA data.

Response: Drought stress imposed at various stages of crop growth resulted in an increase of oxidative stress that causes considerable cellular membrane damage. The extent of damage to membranes was reflected by two main indicators i.e. cellular membrane stability index (MSI%) and lipid peroxidation (Accumulation of MDA). Both these parameter are most effective approach for quantifying the level of plant water stress. In the present work we quantify the cellular membrane stability index instead of lipid peroxidation as a consistent and good parameter indicating the cellular membrane damage in response to water stress. The data for MSI was included in Manuscript.

L 208, 209: incomplete sentence, please edit it: The observation recorded for leaf chlorophyll content was also in the same line 209 differing among the genotypes subjected to drought stress.

Response: Sentence is reframed as reviewer’s suggestion 

L 212: Total phenol content that directly linked with the onion pungency and found to be elevated in response to drought stress: and should be was

Response: Sentence modified as per suggestion

L258: needs to be place before and after respectively.

Response: Changes made in revised MS

Fig 3A-F, Fig. 5: font size is too small and is not readable.

Response: Figures are changed in revised submission 

L 305-353: data mentioned in the result section should be properly checked as at some point the fold expression has not been mentioned. Authors should uniformly mention the fold expression, where it upregulated or down regulated

Response: Fold changes were mentioned in revised MS.

L 354-359: The real time expression in the result section is written as MM part and result is missing. Pl write the result part properly. Move this write up in MM.

Response: Result part is modified in revised MS as per reviewer’s suggestion 

Prediction of SSR data does not fit relevant.

Response: Changes made in revised MS

The authors are advised to correct the MS thoroughly.

Response: All advised corrections are made in revised MS, also the manuscript copyedited from "Scholarly Editing and Translation Services Pvt. Ltd." professional services for language usage, spelling and grammar.

---

## [Decision Letter · Decision Letter 1]

28 Jul 2020

Comparative transcriptome analyses in contrasting onion (Allium cepa L.) genotypes for drought stress

PONE-D-20-06651R1

Dear Dr. G,

We’re pleased to inform you that your manuscript has been judged scientifically suitable for publication and will be formally accepted for publication once it meets all outstanding technical requirements.

Kind regards,

Kundan Kumar, PhD

Academic Editor

PLOS ONE

Reviewers' comments:

Reviewer's Responses to Questions

**Comments to the Author**

1. If the authors have adequately addressed your comments raised in a previous round of review and you feel that this manuscript is now acceptable for publication, you may indicate that here to bypass the “Comments to the Author” section, enter your conflict of interest statement in the “Confidential to Editor” section, and submit your "Accept" recommendation.

Reviewer #1: All comments have been addressed

Reviewer #2: All comments have been addressed

2. Is the manuscript technically sound, and do the data support the conclusions?

Reviewer #1: Yes

Reviewer #2: Yes

3. Has the statistical analysis been performed appropriately and rigorously? 

Reviewer #1: Yes

Reviewer #2: Yes

4. Have the authors made all data underlying the findings in their manuscript fully available?

Reviewer #1: Yes

Reviewer #2: Yes

5. Is the manuscript presented in an intelligible fashion and written in standard English?

Reviewer #1: Yes

Reviewer #2: Yes

6. Review Comments to the Author

Reviewer #1: Authors have responded well to the comments, and incorporated the suggestions in the manuscript.

So, the manuscript appears to be sound now and may be accepted for publication.

Reviewer #2: Authors has made all the correction properly of my comments, and now it can be accepted for the publication

7. PLOS authors have the option to publish the peer review history of their article (what does this mean?). If published, this will include your full peer review and any attached files.

Reviewer #1: **Yes: **Ravi Shankar Singh, Ph.D.

Reviewer #2: No

---

## [Editor Report · Acceptance letter]

30 Jul 2020

PONE-D-20-06651R1 

Comparative transcriptome analyses in contrasting onion (Allium cepa L.) genotypes for drought stress 

Dear Dr. Ghodke:

I'm pleased to inform you that your manuscript has been deemed suitable for publication in PLOS ONE. Congratulations! Your manuscript is now with our production department. 

Kind regards, 

on behalf of

Dr. Kundan Kumar 

Academic Editor

PLOS ONE